MLSTar: automatic multilocus sequence typing of bacterial genomes in R

http://orcid.org/0000-0003-0910-6568 Ferrés Ignacio 1
http://orcid.org/0000-0002-6516-3404 Iraola Gregorio 1 2 giraola@pasteur.edu.uy
1 Bioinformatics Unit, Institut Pasteur de Montevideo , Montevideo , Uruguay
2 Center for Integrative Biology, Universidad Mayor , Santiago de Chile , Chile
Crandall Keith
Electronic publication date: 2018 Jun 15
Publication date: 2018
Volume: 6
Electronic Location ID: e5098
Received 2018 Mar 6; Accepted 2018 Jun 5
Copyright: © 2018 Ferrés and Iraola
Copyright year: 2018
Copyright holder: Ferrés and Iraola
License: This is an open access article distributed under the terms of the Creative Commons Attribution License, which permits unrestricted use, distribution, reproduction and adaptation in any medium and for any purpose provided that it is properly attributed. For attribution, the original author(s), title, publication source (PeerJ) and either DOI or URL of the article must be cited.
License URL: https://creativecommons.org/licenses/by/4.0/

Keywords: MLST, PubMLST, R package, Microbial genomics, Multilocus genotyping, Bacterial genomes

Funding: Agencia Nacional de Investigación e Innovación of Uruguay post-graduation program No. POS_NAC_1_2016_131079 This work was supported by the Agencia Nacional de Investigación e Innovación of Uruguay post-graduation program (No. POS_NAC_1_2016_131079) granted to Ignacio Ferrés. The funders had no role in study design, data collection and analysis, decision to publish, or preparation of the manuscript.

==============================
Multilocus sequence typing (MLST) is a standard tool in population genetics and bacterial epidemiology that assesses the genetic variation present in a reduced number of housekeeping genes (typically seven) along the genome. This methodology assigns arbitrary integer identifiers to genetic variations at these loci which allows us to efficiently compare bacterial isolates using allele-based methods. Now, the increasing availability of whole-genome sequences for hundreds to thousands of strains from the same bacterial species has allowed us to apply and extend MLST schemes by automatic extraction of allele information from the genomes. The PubMLST database is the most comprehensive resource of described schemes available for a wide variety of species. Here we present MLSTar as the first R package that allows us to (i) connect with the PubMLST database to select a target scheme, (ii) screen a desired set of genomes to assign alleles and sequence types, and (iii) interact with other widely used R packages to analyze and produce graphical representations of the data. We applied MLSTar to analyze more than 2,500 bacterial genomes from different species, showing great accuracy, and comparable performance with previously published command-line tools. MLSTar can be freely downloaded from http://github.com/iferres/MLSTar.

Introduction

Multilocus sequence typing (MLST) was introduced in 1998 as a portable tool for studying epidemiological dynamics and population structure of bacterial pathogens based on PCR amplification and capillary sequencing of housekeeping gene fragments (Maiden et al., 1998). In most MLST schemes, seven loci are indexed with arbitrary and unique allele numbers that are combined into an allelic profile or sequence type (ST) to efficiently summarize genetic variability along the genome. Rapidly, MLST demonstrated enhanced reproducibility and convenience in comparison with previous methods such as multilocus enzyme electrophoresis or pulsed-field gel electrophoresis, allowing us to perform global epidemiology and surveillance studies (Urwin & Maiden, 2003). For example, MLST has been applied to elucidate the global epidemiology of Burkholderia multivorans in cystic fibrosis patients (Baldwin et al., 2008) or to understand the dissemination of antibiotic-resistant enterobacteria (Castanheira et al., 2011). However, as MLST started to be massively applied two main drawbacks were uncovered: (i) the impossibility of establishing a single universal MLST scheme applicable to all bacteria; and (ii) the lack of high resolution of seven-locus MLST schemes required for some purposes.

These problems pushed the development of improved alternatives to the original methodology. The extended MLST approach which is based on the analysis of longer gene fragments (Chen et al., 2011) or increased number of loci (Dingle et al., 2008; Crisafulli et al., 2013) proved to improve resolution, and the scheme based on 53 ribosomal protein genes (rMLST) was proposed as an universal approach since these loci are conserved in all bacteria (Jolley et al., 2012). Beyond these improvements, the advent of high-throughput sequencing and the increasing availability of hundreds to thousands whole-genome sequences (WGS) for many bacterial pathogens caused a paradigmatic change in clinical microbiology, making it possible to use nearly complete genomic sequences to enhance typing resolution. This revolution allowed the transition from standard MLST schemes testing a handful of genes to core genome approaches that scaled to hundreds of loci common to a set of bacterial genomes (Maiden et al., 2013).

The generation of this massive amount of genetic information required the accompanying development of database resources to effectively organize and store typing schemes and allele definitions. Rapidly, the PubMLST database (http://pubmlst.org) turned into the most comprehensive and standard resource storing today schemes and allelic definitions for more than 100 microorganisms. Subsequently, the shift to WGS motivated the development of the Bacterial Isolate Genome Sequence Database (BIGSdb) (Jolley & Maiden, 2010), which now encompasses all the software functionalities used for the PubMLST. Also, many tools for automatic MLST analysis from WGS have been developed using web servers like MLST-OGE (Larsen et al., 2012) or EnteroBase (http://enterobase.warwick.ac.uk), pay-walled tools like BioNumerics or SeqSphere+, and open source tools like mlst (http://github.com/tseemann/mlst) or MLSTcheck (Page, Taylor & Keane, 2016). Here, we present MLSTar as the first tool for automatic MLST of bacterial genomes written in R (R Development Core Team, 2008), allowing us to expand the application of MLST tools within this very popular and useful environment for data analysis and visualization.

Methods

Implementation

MLSTar is written in R and contains all data processing steps and command line parameters to call external dependencies wrapped in the package. MLSTar depends on BLAST+ (Camacho et al., 2009) that is used as sequence search engine, and must be installed locally. MLSTar is designed to work on Unix-based operating systems and is distributed as an open source software (MIT license) stored in GitHub (http://github.com/iferres/MLSTar). MLSTar contains four main functions that (i) takes genome assemblies or predicted genes in FASTA format from any number of strains, (ii) performs sequence typing using a previously selected scheme from PubMLST, and (iii) applies standard phylogenetic approaches to analyze the data. An overview of the overall workflow has been outlined in Fig. 1.

Figure 1 Main steps in MLSTar workflow.

Interaction with PubMLST

First step in MLSTar workflow involves to interact with the PubMLST database to select a target scheme. This interaction requires Internet connection because is performed using the RESTful web application programming interface provided by PubMLST. The listPubmlst_orgs() function allows us to list the names of all microorganisms that have any scheme stored in PubMLST. Then, as some microorganisms have more than one scheme (i.e., one classical seven-loci and one core genome scheme), the listPubmlst_schemes() function lists the available schemes for any selected species. Additionally, MLSTar is not restricted only to the MLST definitions present in PubMLST since schemes stored in other databases can be manually downloaded and analyzed with MLSTar.

Calling and storing alleles and sequence types

MLSTar make allele and ST calls from FASTA files containing closed genomes or contigs using BLAST+ blastn comparisons implemented by the doMLST() function. Parallelization is available as internally implemented in R by the parallel package. Also, the doMLST() function can be run at the same time for different schemes using internal R functions like lapply(). Results are stored in a S3 class object named mlst that contains two data.frame objects: one containing allele and ST assignments for the analyzed genomes (unknown alleles or STs are labeled as “u”), and the other storing known allele profiles for the selected scheme. If required, nucleotide sequences for known or novel alleles can be written as multi FASTA files for downstream analyses.

Post analysis

Allele profiles are frequently used to reconstruct phylogenetic relationships among strains. Function plot.mlst() directly takes the mlst class object to compute distances assuming no relationships between allele numbers, so each locus difference is treated equally. Then, identical isolates have a distance of 0; those with no alleles in common have a distance of 1 and, for example, in a seven-loci scheme two strains with five differences would have a distance of 0.71 (5/7). The resulting distance matrix is used to build a minimum spanning tree using igraph (Csardi & Nepusz, 2006) that returns an object of class igraph or a neighbor-joining tree as implemented in APE package (Paradis, Claude & Strimmer, 2004) that returns an object of class phylo. The package also contains a specific method defined as plot.mlst that recognizes the mlst class object and plots the results using the generic plot() function. Additionally, a better resolution analysis based on the variability of the underlying sequences using more sophisticated Maximum-Likelihood or Bayesian phylogenies, can be achieved externally by aligning the allele sequences that are automatically retrieved by MLSTar.

Results and Discussion

Comparison with capillary sequencing data

Multilocus sequence typing analysis based on capillary sequencing has been considered as the gold standard. Hence, we used a previously reported dataset (Page et al., 2017) consisting in 72 Salmonella samples originally tested by capillary sequencing and deposited in the EnteroBase (Alikhan et al., 2018), that were posteriorly whole-genome sequenced. This dataset covers a wide host range and isolation dates of Salmonella strains comprising 32 different STs (Table S1). In average, MLSTar assignments at ST level matched in 92% of cases when compared with capillary sequencing. Additionally, ST calls for five samples that were distinct between capillary sequencing and genome-derived inferences using several software tools (Page et al., 2017), were also discordant in the same way when using MLSTar. This is expected since capillary sequencing is not error free (Liu et al., 2012), in spite of being considered as the gold standard. By the contrary, the result for sample 139K matched between capillary sequencing and MLSTar but most other software tools, except stringMLST (Gupta, Jordan & Rishishwar, 2016), failed to assign confident STs. MLSTar results on the same dataset but in comparison with other softwares designed to screen whole-genome assemblies such as mlst (http://github.com/tseemann/mlst) and MLSTcheck (Page, Taylor & Keane, 2016) matched in 89% and 92% of cases, respectively. These results demonstrate that MLSTar and other software have comparable performance when testing against standard MLST results based on capillary sequencing.

Comparison against BIGSdb

We retrieved 2,726 genomes from the BIGSdb belonging to 10 species most of which are very well-known pathogens (Table S2). For these datasets, reference allele, and ST assignments based on the corresponding standard MLST schemes were extracted from the BIGSdb and compared with results obtained running MLSTar. The concordance at allele and ST levels is shown in Table 1, measured as the percentage of identical assignments between BIGSdb and MLSTar. In average, assignments were 97.9% (SD = 1.95) and 95.6% (SD = 2.5) coincident for alleles and STs, respectively. These results evidence a very good performance of MLSTar in comparison with the reference assignments from the BIGSdb. Additionally, we tested MLSTar using the ribosomal MLST scheme (Jolley et al., 2012) over the same 354 genomes belonging to Staphylococcus aureus and Streptococcus agalactiae. This scheme was conceived as an universal approach for discrimination of bacterial species. Accordingly, the automatic phylogenetic analysis implemented in MLSTar was able to discriminate both species using ribosomal alleles (Fig. 2).

Table 1 Accuracy of MLSTar against reference alleles and STs obtained from BIGSdb, measured as the percentage of correct calls in seven-locus MLST schemes from 11 different pathogens comprising a total of 3,021 genomes.

Species	Genomes	Scheme	
Bordetella spp.	66	adk	fumC	glyA	tyrB	icd	pepA	pgm	ST	
		96.7	96.7	96.7	96.7	96.7	95	96.7	95	
Staphylococcus aureus	72	gdh	gyd	pstS	gki	aroE	xpt	yqiL	ST	
		94.4	94.4	94.5	95.3	94.4	95.2	99.4	93.1	
Helicobacter pylori	79	atpA	efp	mutY	ppa	trpC	ureI	yphC	ST	
		97.5	96.2	98.7	97.5	98.7	97.5	97.5	93.7	
Bacillus cereus	115	glp	gmk	ilv	pta	pur	pyc	tpi	ST	
		98.3	100	100	100	100	96.5	98.2	93.9	
Campylobacter jejuni/coli	176	aspA	glnA	gltA	glyA	pgm	tkt	uncA	ST	
		100	99	100	100	100	100	100	99	
Burkholderia pseudomallei	225	ace	gltB	gmhD	lepA	lipA	narK	ndh	ST	
		98.7	96	93	96	96.9	95.6	96	93	
Streptococcus agalactiae	258	adhP	pheS	atr	glnA	sdhA	glcK	tkt	ST	
		99.2	99.6	99.2	99.2	99.2	99.6	99.6	98.1	
Klebsiella pneumoniae	284	gapA	infB	mdh	pgi	phoE	rpoB	tonB	ST	
		100	100	100	100	100	100	100	100	
Pseudomonas aeruginosa	604	acs	aro	gua	mut	nuo	pps	trp	ST	
		96.4	98.8	98.1	98.3	98.1	98.3	98.8	95.9	
Acinetobacter baumannii	847	cpn60	fusA	gltA	pyrG	recA	rplB	rpoB	ST	
		98.6	97.4	99.3	99.2	97.3	99.1	98.7	94.9	

Figure 2 Phylogeny based on ribosomal alleles.

Staphylococcus aureus (red) and Streptococcus agalactiae (blue) genomes from the BIGSdb (n = 356) were characterized using the universal rMLST scheme (based on 53 ribosomal genes). The phylogenetic tree was automatically generated with the plot.mlst() function using the Neighbor-Joining algorithm from a distance matrix obtained from allele patterns.

Comparison with MLST schemes of close species

The PubMLST database stores schemes for 10 different species within the genus Campylobacter, hence we used this case as negative control to test the specificity of MLSTar. We chose the 172-C. jejuni/coli dataset from BIGSdb and 150 randomly selected C. fetus genomes from a previously published study (Iraola et al., 2017) to run MLSTar against the schemes defined for the remaining Campylobacter species, in order to detect potential false positive calls when analyzing closely related taxa. False positives at both allele and ST levels were not detected neither for C. jejuni/coli nor for C. fetus against the rest (Table S3), indicating that MLSTar is highly specific when working with genetically related bacteria.

Comparison of variable coverage depths and number of genomes

Variable depths of sequencing coverage have been shown to affect the accuracy of different softwares to achieve confident ST calls. In general, most softwares require over than 10× to ensure optimal performance (Page et al., 2017). Here, we tested MLSTar by sampling reads at gradual depths from 10 genomes (representing different STs) from the Salmonella dataset and measured the percentage of correctly assigned STs. Figure 3A shows that MLSTar produce good-enough results when sequencing depth is greater than 10×, and its performance is comparable to similar tools such as MLSTcheck and mlst. Considering that nowadays bacterial genome sequencing experiments typically ensure at least 30× of coverage depth, our results evidence that MLSTar is appropriate for analyzing WGS with average or even slightly lower coverage depths. Additionally, we used a random set of genomes (n = 400) from the BIGSdb dataset to compare the running time between MLSTar, MLSTcheck, and mlst softwares in a single AMD Opteron 2.1 GHz processor, by gradually increasing the number of analyzed genomes from 2 to 400 (Fig. 3B). These results showed that MLSTar is 26-fold faster than MLSTcheck but is threefold slower than mlst (Table S4).

Figure 3 Comparison of MLSTar performance.

(A) Comparison of MLSTar, MLSTcheck and mlst softwares using a dataset of 10 Salmonella genomes de novo assembled at variable coverage depths. (B) Comparison of MLSTar, MLSTcheck, and mlst running times on a single CPU using increasing number of genomes.

Conclusion

The advent of WGS has now allowed to type bacterial strains directly from their whole genomes avoiding to repeat tedious PCR amplifications and fragment capillary sequencing for multiple loci. Today MLST is a valid tool which is frequently used as a first-glimpse approach to explore genetic diversity and structure within huge bacterial population sequencing projects. This incessant availability of genomic information has motivated a constant effort to develop efficient analytical tools from multilocus typing data (Page et al., 2017). Here, we developed a new software package called MLSTar that expands the possibilities of performing allele-based genetic characterization within the R environment. We demonstrate that MLSTar has comparable performance with previously validated software tools and can be applied to analyze hundreds of genomes in a reasonable time.

Supplemental Information

Supplemental Information 1 Table S1.

Dataset containing 72 Salmonella strains originally typed using capillary sequencing and then typed using their whole-genome sequences with several available softwares.

Click here for additional data file.

Supplemental Information 2 Table S2.

Description of genomic datasets downloaded from the BIGSdb and typed with MLSTar.

Click here for additional data file.

Supplemental Information 3 Table S3.

Comparison of MLSTar using closely related species.

Click here for additional data file.

Supplemental Information 4 Table S4.

Running time analysis of MLSTar in comparison with other softwares.

Click here for additional data file.

We thank Daniela Costa and Cecilia Nieves for testing MLSTar.

Additional Information and Declarations

Competing Interests

Author Contributions

Data Availability

The authors declare that they have no competing interests.

Ignacio Ferrés conceived and designed the experiments, performed the experiments, analyzed the data, contributed reagents/materials/analysis tools, prepared figures and/or tables, authored or reviewed drafts of the paper, approved the final draft.

Gregorio Iraola conceived and designed the experiments, performed the experiments, analyzed the data, contributed reagents/materials/analysis tools, prepared figures and/or tables, authored or reviewed drafts of the paper, approved the final draft.

The following information was supplied regarding data availability:

Github: https://github.com/iferres/MLSTar.

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
