# Peer review of "MLSTar: automatic multilocus sequence typing of bacterial genomes in R"

_PeerJ, doi:10.7717/peerj.5098_

## Round 0.1 · original submission · Major Revisions

I have now received two reviews and one includes attachments (be sure you get those). The reviewers require some additional work and one reviewer provides some details and references for doing more comparisons of your approach with others available. I hope you can accommodate these reviews successfully.

·

Basic reporting

Excellent standard of English and a good literature review.

It would be nice to include a sentance or 2 in the introduction motivating why MLST is so useful, such as outbreak investigaitons, survellance etc...

You might find the following preprint useful:

GrapeTree: Visualization of core genomic relationships among 100,000 bacterial pathogens
https://www.biorxiv.org/content/early/2017/11/09/216788

Experimental design

Whilst Minimum spanning trees are fast, the quality of tree isnt the best, particularly from alleles. You can get better resolution using the underlying sequences (so based on SNPs) and ML based methods.

The performance comparisons are weak. Just testing it on C. coli doesnt tell us much, particularly when nearly 100 of the samples have the same ST, and the actual alleles of the rest are are drawn from a tiny pool. Try testing it on a few different organisms, with diversity, for example the 10 most common pathogens.

You should also add in controls so that you can calculate specificity and sensitivity. There are 10 different campy MLST schemes, does your software falsely return an ST if you test your 400 genomes against the other 9 schemes?

The numbers in Table 1 need to be double checked. The accuracy of the ST should be less than or equal to the lowest accuracy for the individual alleles, with a lower bound of multiplying each of those numbers (~97%).

Validity of the findings

The software appears to call STs when there are only partial matches. For MLST to be called properly there must be a 100% match in all alleles.

Whilst extended MLST schemes are mentioned in the title, there is no validation or results in the paper itself. Does it actually work on extended schemes or is this theoretical? Either include some results from an extended scheme or remove this claim.

Some MLST schemes have alleles completely missing, how does your method deal with this?

How does your software compare against the data in this MLST comparison paper (which you have cited)?
http://mgen.microbiologyresearch.org/content/journal/mgen/10.1099/mgen.0.000124

Reviewer 2 ·

Basic reporting

MLSTar software is tested and validated to carry out MLST (~7 housekeeping genes) so discussion about cgMLST should be removed from the abstract.


README.md document should state that script works only on the Unix-based operating system.

Experimental design

'no comment'

Validity of the findings

please see attached document

Additional comments

Does MLSTar software carry out cgMLST ? Would there be any future development for MLSTar to carry out cgMLST?

Annotated reviews are not available for download in order to protect the identity of reviewers who chose to remain anonymous.

---

## Round 0.2 · accepted · Accept

Thank you for your careful revision and accommodation of previous critiques. I have sent your paper back to the harsher of the two previous reviewers and this reviewer now finds your paper acceptable, as do I. Congratulations.

# ·

Basic reporting

Excellent

Experimental design

Excellent

Validity of the findings

Thank you for including much more comprehensive experiments.